# Leucocyte Abnormalities in Synovial Fluid of Degenerative and Inflammatory Arthropathies

**DOI:** 10.3390/ijms24065450

**Published:** 2023-03-13

**Authors:** Chiara Baggio, Roberto Luisetto, Carlotta Boscaro, Anna Scanu, Roberta Ramonda, Mattia Albiero, Paolo Sfriso, Francesca Oliviero

**Affiliations:** 1Rheumatology Unit, Department of Medicine—DIMED, University of Padova, 35128 Padova, Italy; 2Department of Surgery, Oncology and Gastroenterology-DISCOG, University of Padova, 35128 Padova, Italy; 3Department of Medicine, University of Padova, 35128 Padova, Italy; 4Veneto Institute of Molecular Medicine, Via G. Orus 2, 35129 Padova, Italy; 5Department of Woman’s and Child’s Health, University of Padova, 35128 Padova, Italy

**Keywords:** synovial fluid, rheumatoid arthritis, osteoarthritis, psoriatic arthritis, crystal-induced arthritis, genomic instability, apoptosis, inflammation

## Abstract

Genome damage has been related to the induction of autoimmune processes, chronic inflammation, and apoptosis. Recent studies suggest that some rheumatological diseases are associated with overall genomic instability in the T cell compartment. However, no data regarding leucocyte abnormalities in synovial fluid (SF) and their relationship with inflammation are available. The aim of this study was to investigate cellular phenotypes in SF collected from patients with different inflammatory arthropathies, including rhematoid arthritis (RA), psoriatic arthritis (PsA), crystal-induced arthritis (CIA), and non-inflammatory arthropathies, such as osteoarthritis (OA). We found high percentage of micronuclei in SF from CIA compared to the other groups and a high frequency of pyknotic cell in RA and CIA patients. A correlation between pyknosis and immature polymorphonuclear cells with local inflammatory indices was observed. The study of the apoptosis process revealed an increased BAX expression in CIA and RA compared to OA and PsA, while Bcl-2 was higher in CIA. Caspase-3 activity was increased in SF from RA patients and correlates with inflammatory and anti-inflammatory cytokines. In conclusion, our results showed that inflammatory SF is associated with genomic instability and abnormal cell subsets.

## 1. Introduction

Rheumatological joint diseases, including rheumatoid arthritis (RA), psoriatic arthritis (PsA), crystal-induced arthritis (CIA), and osteoarthritis (OA), are chronic conditions affecting a large proportion of the population, and they represent a leading cause of disability and comorbidities worldwide [1]. Although the risk factors and the pathogenetic mechanisms related to these diseases are different, they share some features, such as local edema, inflammatory reactions, and articular damage. Abundant effusions are often presented in these patients, and synovial fluid (SF) may provide important information on cellular and humoral changes occurring during the inflammatory process.

SF leucocytes increase during an acute phase of disease, reaching thousands of different cells per mmc [2]. The majority of the cells in non-inflammatory SF are represented by mononuclear cells (i.e., monocytes), while a high percentage of neutrophils are observed in inflammatory SF that correlate with the number of SF white blood cell (WBC).

It is known that leucocytes can present abnormalities depending on cell state, aging, or upon mutations [3]. These abnormalities include nuclear defects, such as polyploid (binucleated) cells, due to an extra copy of the genome sequestered in a separated nucleus, and micronuclei (MN), small nuclei in nuclear proximity containing intact or fragments of chromosomes [4]. Other abnormal leucocyte morphologies include hypo- and hyper-segmentation and immature forms.

It has been hypothesized that genome damage might be related to the induction of autoimmune and chronic inflammatory diseases [5]. Self-DNA released from MN, for instance, may be detected by DNA-sensors and trigger an innate autoimmune response and chronic inflammation through the release of interferon type I cytokines [6]. Inflammation can also induce MN which, in turn, sustain the pathological process [5].

Genomic instability can lead to altered gene expression and induce cell apoptosis. Apoptosis can affect nuclear integrity disturbing, both the nuclear lamina and envelope, and resulting in cell death [7].

Recent studies suggest that some rheumatological diseases, including rheumatoid arthritis (RA), are associated with overall genomic instability in the T cell compartment [8] and increased sensitivity to DNA damage [9]. However, no data regarding leucocyte abnormalities in synovial fluid (SF) and their relationship with inflammation are available. The aim of the study is to investigate this aspect in different SF phenotypes collected from patients with different inflammatory and non-inflammatory arthropathies.

## 2. Results

### 2.1. Synovial Fluid Characteristics

SF characteristics are shown in Table 1. All SFs were collected during an active phase of the disease, and, except for OA, all WBC were >2000 cell/mm^3^. The most inflammatory degree, in terms of both WBC count and PMN percentage, was found in SF collected from CIA, followed by RA and PsA. SF from OA patients showed typical non-inflammatory features. Patients with PsA were the youngest among the groups.

### 2.2. Cytogenic Evaluation of SF Leucocytes

All the preparations were examined for MN, NA (nuclear abnormalities), and karyolitic, karyorrhectic, and pyknotic cells. The morphologies of the different cell types are illustrated in Figure 1, and the results are summarized in Table 2.

The hypersegmented neutrophils band, hyposegmented (immature PMN), and hypersegmented PMN were higher in SF from patients with RA, PsA, and CIA compared to those with OA. The percentage of MN was significantly higher in CIA patients than in the OA patients, but there were no statistically significant differences in the MN frequency between the other groups of patients. Binucleated monocytes were higher in OA patients with respect to those with CIA. No significant differences were found in the frequency of M and PMN with vacuoles between the patient groups analyzed.

### 2.3. MN Frequencies in SF Leukocytes

MN and other nuclear abnormalities are considered biomarkers of genotoxic events and chromosomal instability. MGG staining was used to detect MN in PMN and M from OA, RA, PsA, and CIA patients. Figure 2 shows a significant increase in MN found in the CIA and PsA cohorts relative to the OA cohort (*p* < 0.05). Data are summarized in Appendix A.

### 2.4. Cell Death Rate in SF Leukocytes

To monitor cytotoxic effects, we evaluated the frequencies of karyorrhexis, karyolysis, and pyknosis in SF leukocytes (Figure 3). Pyknotic cell frequency was significantly higher in RA (*p* < 0.05) and CIA (*p* < 0.01) patients than in OA patients. The percentage of karyorrhectic and karyolitic cells were higher in RA patients with respect to OA and PsA patients (*p* < 0.05), although significance was reached only for karyorrhexis (Figure 3B,C). A significant difference was found in the rate of cell death between RA, CIA, and OA patients (*p* < 0.01) (Figure 3D) (Appendix A).

### 2.5. Cytokines

SF cytokine, chemokine, and growth factor levels are presented in Table 3. The non-parametric analysis of variance indicated a significant difference between groups for all the cytokines considered. The levels of IL-1β were higher in the SF of CIA and RA patients with respect to PsA and OA. CIA showed the highest levels of IL-6 and TGF, while SF from RA patients had the highest concentrations of IL-8, IL-10, and TNFα. Although higher in CIA, the levels of TGFβ in SFs did not differ among groups.

### 2.6. Association between Abnormal Populations and Inflammation

A strong positive correlation between local inflammatory cellular indices including WBC and PMN and the cytokines IL-1β, IL-6, IL-8, and IL-10 was found (Table 4).

Immature PMN percentage was positively correlated with WBC count and PMN percentage and with the levels of IL-1β, IL-6, IL-8, and IL-10; it was negatively correlated with M and L percentage (Table 4). The frequency of MN was positively correlated with IL-8. Pyknosis was positively correlated with WBC, PMN, and with the levels of IL-1β, IL-8, and IL-10, and negatively with M and L. The percentage of karyolitic cells was positively associated with WBC and with the levels of IL-1β, IL-8, IL-10, TNF, and TGFβ. Finally, binucleated monocytes were positively correlated with M and L, and were negatively correlated with WBC, PMN, IL-1β, IL-6, and IL-10 (Table 5).

### 2.7. Analysis of Apoptotic Genes in SF Leukocytes

The study of gene expression revealed a significant difference in BAX and Bcl-2 mRNA in the four group of diseases, as evidenced by the non-parametric analysis of variance (BAX *p* = 0.02, Bcl-2 *p* = 0.05). More specifically, BAX expression was increased in CIA and RA compared to OA and PsA (Figure 4A) while Bcl-2 was higher in CIA, followed by PsA, RA, and OA (Figure 4B). No differences were observed in BAK expression and in the proapoptotic regulators BAD and BID between the patient groups analyzed (Figure 4C–E) (Appendix A). To support PCR analysis, we evaluated differential expression of Bax and Bcl-2 at the protein level in the four groups of diseases. Although non-significant, the levels of BAX transcripts and protein products showed the same trend, while Bcl-2 protein levels were higher in RA patients with respect to the other groups of diseases (Appendix A).

### 2.8. Caspase-3 Activity in SFs and Correlations with Local Inflammatory Indices

Analysis of caspase-3 revealed that RA patients had increased levels of activity for this enzyme compared to the other diseases, and they were significant when compared to OA and PsA patients (*p* < 0.05) (Figure 5) (Appendix A). As regards the association with local inflammation, we found a positive correlation between caspase-3 activity and WBC and PMN, and with the levels of IL-1β, IL-8, and IL-10 (Table 6). By contrast, there was a negative correlation between caspase-3 activity and the percentage of L (Table 6).

## 3. Discussion

The aim of the study was to investigate the relationship between the presence of genomic instability and the degree of inflammation in the SF obtained from RA, PsA, CIA, and OA patients. Although genomic instability has been associated with both inflammatory and autoimmune diseases, there are no data in the literature describing abnormalities in SF leucocytes from chronic rheumatological conditions.

The SF collected from patients with the inflammatory diseases RA, PsA, and CIA had classical inflammatory features, with CIA displaying a higher number of WBCs and PMN. OA, which is considered a low-grade inflammatory disease, showed non inflammatory cellular aspects (Table 1). In the whole population, local inflammatory cellular indices correlated positively with the levels of the most important cytokines involved in the pathophysiology of these diseases, including IL-1β, IL-6, IL-8, and IL-10.

To investigate signs of genotoxicity/cytotoxicity, we searched for the following cytological abnormalities: (1) nuclear abnormalities (MN, nucleoplasmic bridges, nuclear buds, fused/circular nuclei), (2) cell death biomarkers (necrotic and apoptotic cells), and (3) biomarkers indicative of cell division rate or cell division failure (binucleated cells) [10]. Micronuclei have been recognized as one of the most important biomarkers for mutagenic and genotoxic damage. They have been associated with different pathological conditions including accelerated ageing and inflammation [10]. It has been demonstrated that MNi are sensed by receptors of the innate immune system to induce inflammation through cGAS-STING cascade activation and its downstream interferon regulatory factor signaling [10]. Interestingly, we found that the percentage of MN was significantly higher in CIA compared to the other groups, and that it was positively correlated with the levels of IL-8. It has been found that molecules, such as reactive oxygen and nitrogen species, and inflammatory cytokines released in SF during the inflammatory process, may induce DNA damage in vitro [11]. The higher levels of MN found in SF from CIA patients might be explained by the genotoxic effect induced by pathogenic MSU or CPP crystals. This hypothesis is supported by the study from Licandro et al. demonstrating that NLRP3 inflammasome activators, such as MSU crystals, directly induce DNA breaks and provoke robust ROS production [12].

Neutrophils are the main actors in acute inflammatory responses. In different inflammatory conditions, immature neutrophils (with non-segmented nuclei) can be released from the bone marrow, causing a so-called left shift [13]. Banded neutrophils are younger than their segmented counterparts. However, the hypersegmented morphology of neutrophils implies increased maturation compared with banded and normal neutrophils [14]. We found higher levels of immature and hypersegmented PMN in RA, PsA, and CIA patients compared to OA patients. In addition, we found that immature PMN percentage was positively correlated with cellular inflammatory indices (WBC, PMN), and with the levels of IL-1β, IL-6, IL-8, and IL-10. It is possible that the massive release of neutrophils during the active phase or after an acute attack of disease leads to depletion of well-functioning neutrophils. Kamp et al. also reported that hypersegmented neutrophils do not only differ in phenotype but also in functionality. For example, endothelial adhesion is decreased in these cells [15], suggesting that this phenotype may be correlated with a descending phase of inflammation.

Cell death biomarkers evaluated in this study were pyknosis, karyolysis, and karyorrhexis. Pyknosis has been widely considered as a marker of cell death, and it is characterized by an irreversible condensation of chromatin. It commonly occurs in both apoptotic and necrotic cell death. Karyolysis is the disintegration and dissolution of the nucleus of a necrotic cell, usually associated with karyorrhexis, while the nucleus of an apoptotic cell usually dissolves after karyorrhexis [16]. We found that pyknotic cell frequency was significantly higher in RA and CIA patients than in non-inflammatory OA patients and was correlated to cellular inflammatory indices and IL-1β, IL-8, and IL-10 levels. In CIA, crystals can affect apoptotic processes in various cells, including neutrophiles, through the release of ROS and NOS, thereby promoting apoptosis [17]. MSU and CPP crystals have also been shown to induce necrosis [18]. The percentage of karyorrhectic and karyolitic cells were higher in SF from RA patients with respect to the samples collected from the other groups and was positively associated with WBC count and the levels of IL-1β, IL-8, IL-10, TNF, and TGFβ. We found that neutrophils were characterized by the greatest frequency of pyknosis, and it is known that PMN actively contributes to tissue damage in inflammatory arthritis [19,20]. Apoptosis is a controlled form of cell death that minimizes the risk of collateral damage to surrounding tissues by leukocytes, in particular PMN [19,21]. Therefore, during the resolution phase of inflammation, both apoptosis and clearance are crucial events, and their defects might contribute to prolonged inflammation [22]. Our data show that SF samples characterized by higher levels of inflammation are also those with a high frequency of pyknosis. This could represent a mechanism that precedes the final stage of resolution of inflammation also mediated by apoptosis.

Apoptosis is a complex and tightly regulated process divided into three phases: initiation, commitment, and execution. The Bcl-2 protein family regulates the commitment phase, while caspases regulate the execution of apoptosis. Caspase-3 is classified as effector caspase and, among all executioner caspases, is the most important [23,24]. Anti-apoptotic members, such as Bcl-2 bind to the pro-apoptotic effectors BAX/BAK, preventing their activation. Diverse stimuli can induce the BH3-only proteins (BID, BIM), which selectively compete for binding with the anti-apoptotic proteins, thus, facilitating the release of BAX/BAK, which consequently leads to the activation of caspases and cell destruction [20,24]. The genetic expression analysis was carried out using total RNA extraction, and it reflects the expression of leukocytes present in the SF. We found that BAX expression was increased in CIA and RA compared to OA and PsA, while Bcl-2 was higher in CIA followed by PsA, RA, and OA. At the protein level, while BAX showed the same trend, Bcl-2 was higher in RA patients. This discrepancy between transcripts and proteins may be attributed to different levels of Bcl-2 transcriptional activation, a delayed protein synthesis, or a short half-live of the protein [25]. Unexpectedly, the levels of the anti-apoptotic factor were higher in SF samples characterized by a greater degree of inflammation. It has been reported that alterations in the expression of Bcl-2 may explain the resistance of leukocytes to apoptosis, resulting in a prolonged production of proinflammatory cytokines [26]. In addition, the microenvironment of SF is characterized by pro-inflammatory factors responsible for the in loco persistence of activated and long-surviving neutrophils [19]. It must be considered that the equilibrium between pro-apoptotic and anti-apoptotic proteins is an essential element for the initiation of apoptosis, and the ratio between these factors is generally used to determine the fate of the cell [24]. Therefore, to better understand the final outcome of the expression of these factors, we evaluated the caspase-3 activity. We found that SF from RA patients had increased levels of caspase-3 activity compared with other diseases, and the levels of caspase-3 activity positively correlate with WBC count, PMN, and with the levels of both pro-inflammatory and anti-inflammatory cytokines including IL-10. Of note, it has been reported that IL-10 triggers apoptosis [27], suggesting that programmed cell death is functional to the resolution of inflammation.

In conclusion, inflammatory SF samples from RA, PsA, and CIA patients are associated with abnormal cell subsets and apoptosis that could be involved in the resolving phase of the inflammatory process. A deeper knowledge of different abnormal cell subsets and their function could lead to a better understanding of the underlying mechanisms for the resolution of inflammation in these diseases.

## 4. Materials and Methods

### 4.1. Reagents

May–Grunwald (63590) and Giemsa staining (1.09204.0500), Bradford (B6916), PBS (P4417), and methanol (179337) were from Sigma-Aldrich (St. Louis, Missouri, USA). ELISA kits for interleukin (IL)-1β (88–7261), IL-6 (88–7066), IL-8 (88–726), IL-10 (88–7106), and TGFβ (88–8350) were from Thermo Fisher Scientific (Waltham, Massachusetts, USA), and TNFα was from BioLegend (San Diego California, USA; 430204). The Total RNA purification kit was from Norgen Biotek (Thorold, Canada; 17200). The SensiFAST™ cDNA Synthesis Kit (BIO-65054) and SensiFAST™ SYBR^®^ Lo-ROX Kit (BIO-94050) were purchased from Bioline (London, UK). Caspase-3 activity was determined using a colorimetric assay kit from Abcam (Cambridge, UK; ab39401).

### 4.2. Study Design

Synovial fluids (SFs) were collected by arthrocentesis from swollen knees of untreated adult patients in the acute stage with RA (n = 6), PsA (n = 12), CIA (n = 10), and OA (n = 7) who attended the outpatients’ clinic of the Rheumatology Unit at Padua University Hospital for acute swelling and were diagnosed according to ACR (R), CASPAR (r) and EULAR criteria [28,29,30,31,32], respectively. Regarding patients with CIA, we included three samples of SF with MSU crystals and seven samples of SF with CPP crystals. Patients’ demographic and SF characteristics are shown in Table 1. Discarded samples were stored and studied under protocols approved by the local institutional review board and with the subjects’ informed consent (approval #39872).

### 4.3. Synovial Fluid Collection

Synovial fluids samples were collected in EDTA and plain tubes and analyzed by ordinary and polarized light microscopy. Total white blood cell (WBC) count was performed using a Bürker counting chamber. Differential leucocyte count was performed by microscopic examination of 300 cells on MG-stained preparations. The percentage of polymorphonuclear neutrophils (PMN), monocytes (M), and lymphocytes (L) in the SF was recorded. Crystal search was performed using compensated polarized light microscopy. Monosodium urate (MSU) and calcium pyrophosphate (CPP) crystals were distinguished by their shape, degree, and sign of birefringence [33]. After examination, SF samples were centrifuged at 800 rpm for 20 min to collect cells and stored at −20 °C until further analysis.

### 4.4. Cellular Morphology and Cytogenic Evaluation of Leucocytes

May–Grunwald–Giemsa (MGG) staining was used for studying cellular morphology, including cytoplasm, granules, vacuoles, and the basement membrane, and to perform a cytogenic evaluation of leucocytes. Air-dried SF smears were fixed with methanol and stained using a stepwise procedure with MGG. The smear was transferred to May–Grunwald solution diluted 1:1 in water for 3 min. Excess stain was drained off on filter paper (without blotting), and the smear was then transferred to dilute 1:9 Giemsa solution for 7 min. The smear was transferred to water, rinsed for 30 s, and then dried. Oil immersion microscopy with 1000× magnification was applied for analysis; for each smear we examined at least 200 leucocytes. Leukocytes were identified on the basis of morphology; in each field the number of cell profiles corresponding to neutrophils, monocytes, and lymphocytes were recorded. All slides were examined for micronuclei (MN) (a micronucleus is defined as a round cytoplasmic inclusion having a diameter one-tenth to one-third that of the primary nucleus), and types of nuclear abnormality (NA) included binucleated cells (contains two nuclei that are not attached and similar in size), and karyolitic, karyorrhectic, and pyknotic cells [34]. Apoptotic morphology was identified according to the typical criteria of cell shrinking, nuclear condensation, and fragmentation (apoptotic bodies). Apoptotic neutrophils were distinguished from apoptotic lymphocytes on the basis of acidophilic or basophilic cytoplasmic staining. Karyorrhectic cells are characterized by fragmentation and eventual disintegration of the nucleus. These cells may be undergoing a late stage of apoptosis, but this has not been conclusively proven. Karyolitic cells stain uniformly with eosin due to the complete dissolution of the chromatin (apparent as a ghost-like image). They may represent a very late stage in the cell death process [28]. Results were expressed as a percentage of total leukocytes counted in a slide.

### 4.5. Cytokine, Chemokine, and Growth Factor Measurement in Synovial Fluid

The following cytokines, chemokines, and growth factors were measured in SF after appropriate PBS dilutions using commercially available enzyme-linked immunosorbent assay (ELISA) kits: interleukin (IL)-1β (dilution 1:3, sensitivity: 2 pg/mL), IL-6 (dilution 1:200, sensitivity: 2 pg/mL), IL-8 (dilution 1:30, sensitivity 2 pg/mL), IL-10 (dilution 1:1, sensitivity 2 pg/mL), TGFβ (dilution 1:30, sensitivity 8 pg/mL) (Thermo Fisher Scientific, Waltham, Massachusetts, United States), and TNFα (dilution 1:1, sensitivity: 7.8 pg/mL; BioLegend).

### 4.6. RNA Extraction and Real-Time qPCR

Selected SF samples (n = 5) from PsA, CIA, RA, and OA patients were centrifuged at 1500 rpm for 10 min to collect cells and stored at −20 °C. RNA was isolated using a Total RNA purification kit (Norgen Biotek) according to the manufacturer’s recommendations and quantified with a NanoDrop™ 2000 Spectrophotometer (Thermo Fisher Scientific). cDNA was synthesized using the SensiFAST™ cDNA Synthesis Kit (Bioline). The expression levels of BCL-2, BAK, BID, BAD, and BAX were measured by real-time quantitative PCR (qPCR) using SensiFAST™ SYBR^®^ Lo-ROX Kit (Bioline) via Quant Studio 5 real-time PCR system (Thermo Fisher Scientific). Target genes were normalized to GAPDH and analyzed using the 2−ΔCt method. The primer sequences are given as follows:
**Gene****Forward Primer****Reverse Primer**BCL-2ATGTGTGTGGAGAGCGTCAAACAGTTCCACAAAGGCATCCBAKTCATCGGGGACGACATCAACCAAACAGGCTGGTGGCAATCBIDCTTGCTCCGTGATGTCTTTCTCCGTTCAGTCCATCCCATTTBADCGGAGGATGAGTGACGAGTTGATGTGGAGCGAAGGTCACTBAXTCTGACGGCAACTTCAACTGTTGAGGAGTCTCACCCAACCGAPDHTGCACCACCAACTGCTTAGCGGCATGGACTGTGGTCATGAG

### 4.7. Western Blot

Leukocytes from synovial fluids were lysed with 50 μL lysis buffer (Abcam, Cambridge, United Kingdom) for 10 min at 4 °C. Supernatants were than collected for SDS-PAGE western blotting and centrifuged at 10,000 rpm for 15 min. Protein quantification was performed using a BCA assay kit (Euroclone, Milan, Italy). Proteins (60 μg) were separated on SDS-PAGE. Membranes were then blocked and probed using the anti-Bax and anti-Bcl-2 mouse primary monoclonal antibodies (Biolegend) and anti-actin mouse primary monoclonal antibodies (Biolegend, San Diego, CA, USA). After washing, membranes were incubated with mouse secondary horseradish peroxidase-conjugated antibodies (Biolegend, San Diego, CA, USA). Bands were detected through chemiluminescence using Western brightTM 143 Quantum (Advansta, Menlo Park, CA, USA). Images were acquired by the Alliance mini HD9 Imaging System (Uvitec, Cambridge, UK). Densitometric analysis of bands was performed using Image J version 1.47 software (National Institutes of Health, Bethesda, MD, USA).

### 4.8. Caspase-3 Activity

The same SF samples used for RNA extraction were tested for caspase-3 activity (n = 5). Caspase-3 activity was determined using a colorimetric assay kit based on the formation of the chromophore p-nitroaniline (p-NA) by cleavage from the labeled substrate DEVD-pNA (Abcam) according to the manufacturers’ instructions. Briefly, cells were lysed in the supplied lysis buffer for 10 min at 4 °C. Supernatants were collected (200 μg of protein in 50 µL) and incubated with the supplied reaction buffer containing dithiothreitol and DEAD-pNA as substrates at 37 °C for 2h. The reactions were measured by changes in absorbance at 405 nm using the Byonoy microplate reader (Byonoy GmbH, Hamburg, Germany).

### 4.9. Statistical Analysis

Data are reported as median and IQR. The Shapiro–Wilk test was used to analyze the distribution of continuous variables. For non-normal distributed data, the Kruskal–Wallis followed by Dunnet post hoc tests were used for multiple comparisons. Categorical variables were compared using the Chi-square test. Spearman correlation analysis was used to determine correlations. Statistical analysis was performed using GraphPad Prism version 8 (GraphPad Software Inc., La Jolla, CA, USA). A *p* value < 0.05 was considered significant.

## Figures and Tables

**Figure 1 ijms-24-05450-f001:**
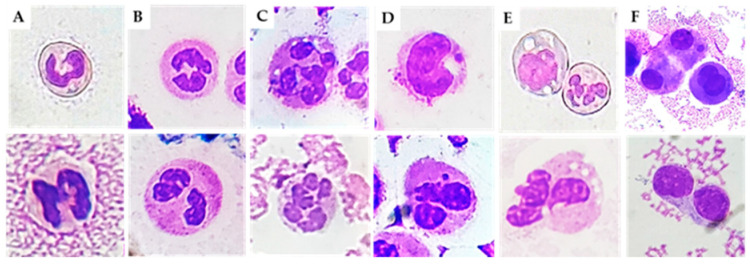
Cytogenic and cellular morphology evaluation of SF leucocytes. Synovial fluid smears were stained using MGG staining. (**A**) Band PMN, (**B**) hyposegmented PMN, (**C**) hypersegmented PMN, (**D**) MN, (**E**) cells with vacuoles, (**F**) binucleated. PMN, polymorphonuclear cells; M, monocytes; L, Lymphocytes; MN, micronucleus.

**Figure 2 ijms-24-05450-f002:**
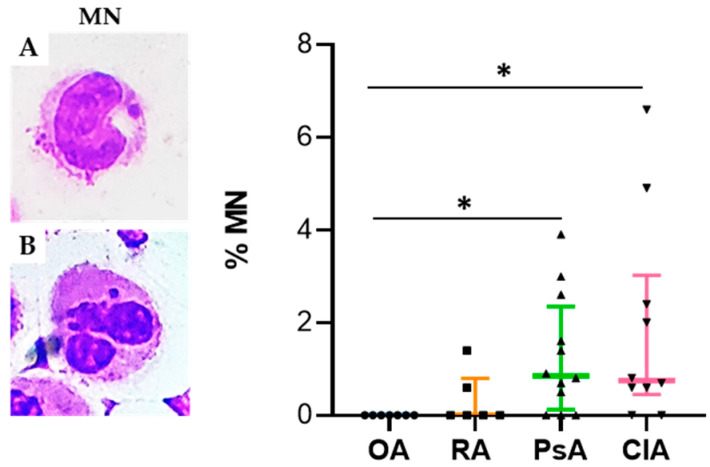
Micronucleus frequencies in SF leukocytes from OA (n = 7), RA (n = 6), PsA (n = 12), and CIA (n = 10). Left panel: representative image of monocytes with MN (**A**) and PMN with MN (**B**) stained with MGG. Right panel: rate of MN in SFs leukocytes. Data are shown as the median (IQR). *p* calculated according to the Kruskal–Wallis test. Dunn’s post hoc test: * *p* < 0.05. Abbreviations are as follows: OA, osteoarthritis; RA, rheumatoid arthritis; PsA, psoriatic arthritis; CIA, crystal-induced arthritis; PMN, polymorphonuclear cells; M, monocytes; L, lymphocytes; MN, micronucleus.

**Figure 3 ijms-24-05450-f003:**
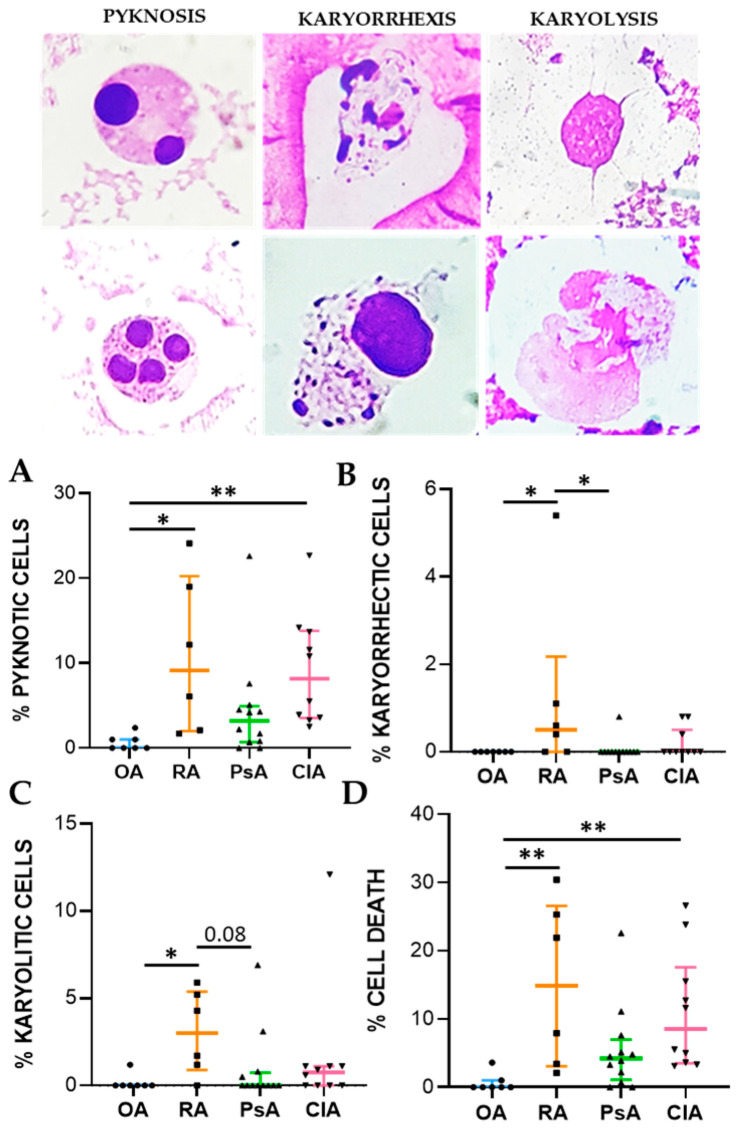
Cell death rates of leukocyte cells in SF from OA (n = 7), RA (n = 6), PsA (n = 12), and CIA patients (n = 10). Upper panel: representative image of pyknotic, karyorrhectic, and karyolitic leukocytes stained with MGG. Lower panel: rate of pyknotic leukocytes (**A**), karyorrhectic leukocytes (**B**), karyolitic leukocytes (**C**), and cell death (**D**). Data are shown as the median (IQR). *p* calculated according to the Kruskal–Wallis test. Dunn’s post hoc test: * *p* < 0.05, ** *p* < 0.01. Abbreviations are as follows: OA, osteoarthritis; RA, rheumatoid arthritis; PsA, psoriatic arthritis; CIA, crystal-induced arthritis.

**Figure 4 ijms-24-05450-f004:**
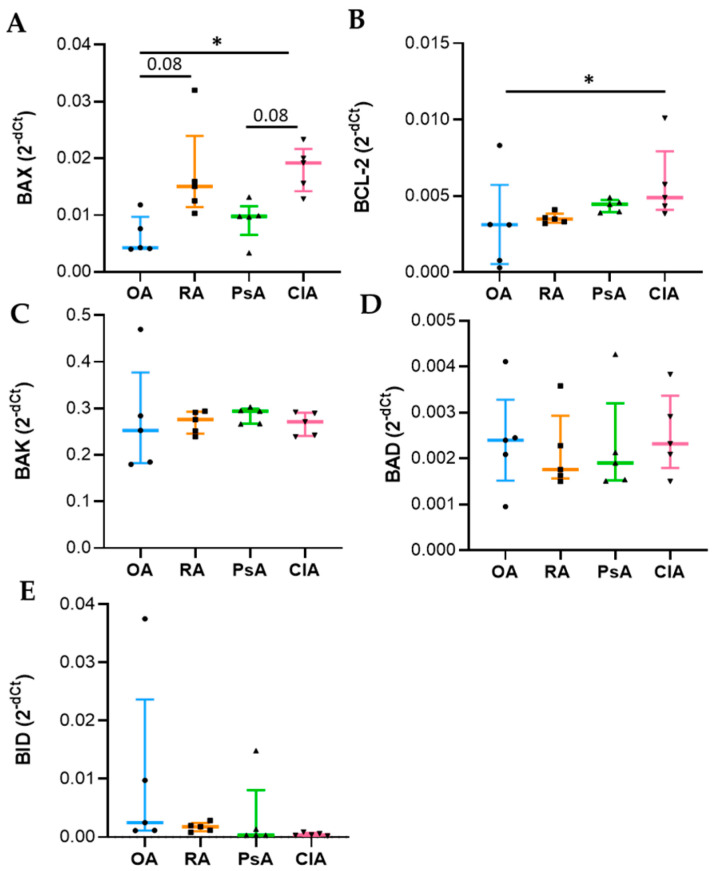
Apoptotic gene expression in SF from OA, RA, PsA, and CIA patients. Measurement of BAX (**A**), BCL-2 (**B**), BAK (**C**), BAD (**D**), and BID (**E**) gene expression was performed in selected SF (n = 5). Data are shown as the median (IQR). *p* calculated according to the Kruskal–Wallis test. Dunn’s post hoc test: * *p* < 0.05. Abbreviations are as follows: OA, osteoarthritis; RA, rheumatoid arthritis; PsA, psoriatic arthritis; CIA, crystal-induced arthritis.

**Figure 5 ijms-24-05450-f005:**
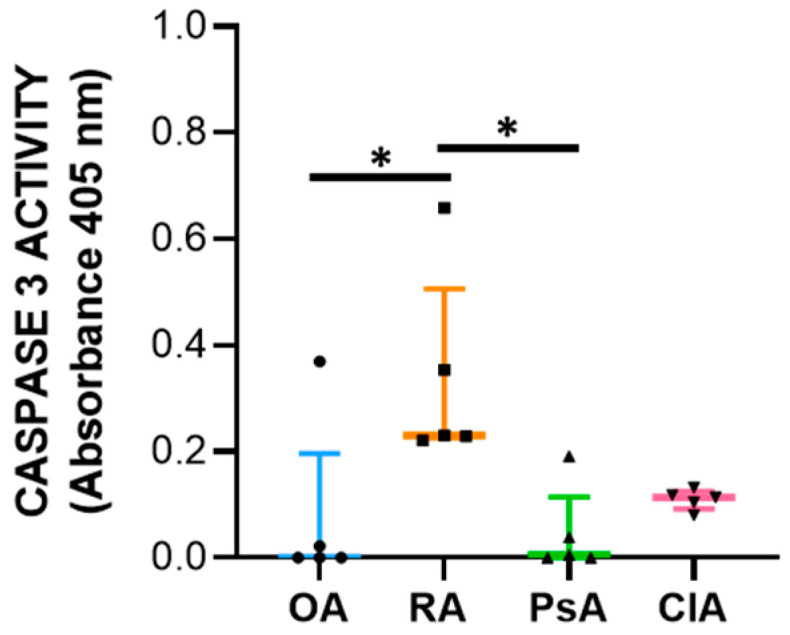
Caspase-3 activity determined in selected SF (n = 5). Data are shown as the median (IQR) of optical density values. *p* is calculated according to the Kruskal–Wallis test. Dunn’s post hoc test: * *p* < 0.05. Abbreviations are as follows: OA, osteoarthritis; RA, rheumatoid arthritis; PsA, psoriatic arthritis; CIA, crystal-induced arthritis.

**Table 1 ijms-24-05450-t001:** Characteristics of the patients included in the study and their synovial fluid.

	OA	RA	PsA	CIA	*p*
Patients, n	7	6	12	10	
Age, years (IQR)	63 (55–69) §	60 (51–70)	50 (26–56)	64 (54–76) §	0.01
Sex, n	F, 6; M, 1	F, 3; M, 3	F, 5; M, 7	F, 4; M, 6	ns
WBC, n/mm^3^ (IQR)	100 (100–200)	12,500 (6250–25,000) *	5600 (3500–12,000) *	17,000 (9875–24,500) *	0.0001
PMN, % (IQR)	3 (0–7)	74 (56–89) *	59 (13–76)	90 (74–95) #	0.0001
M, % (IQR)	70 (49–76)	17 (10–28) *	29 (17–61) °	8 (4–17) #	0.0004
L, % (IQR)	25 (21–44)	7 (2–19)	13 (5–30) °	2 (0–5) #	0.002

Data are expressed as the median and interquartile range (IQR). Categorical variables were compared using the χ^2^ test; § *p* < 0.05 vs. PsA. *p* calculated according Kruskal–Wallis test. Dunn’s post hoc test: * *p* < 0.05 vs. OA, # *p* < 0.001 vs. OA, ° *p* < 0.05 vs. CIA. Abbreviations are as follows: OA, osteoarthritis; RA, rheumatoid arthritis; PsA, psoriatic arthritis; CIA, crystal-induced arthritis; WBC, white blood cell; PMN, polymorphonuclear cells; M, monocytes; L, lymphocytes; IQR, interquartile range.

**Table 2 ijms-24-05450-t002:** Cytogenic and cellular morphology evaluation of PMN and M in SF from OA (n = 7), RA (n = 6), PsA (n = 12) and CIA (n = 10) patients.

		OA	RA	PsA	CIA	*p*
PMN	Band, % (IQR)	0.0 (0.0–1.0)	5.2 (2.3–8.3) *	2.1 (0.0–4.1)	4.1 (3.2–4.8) *	0.002
	Hyposegmented, % (IQR)	0.0 (0.0–1.3)	9.2 (5.8–12.0) °	4.7 (1.6–9.9) °	9.5 (5.9–13) °	0.001
	Hypersegmented, % (IQR)	0.0	1.9 (1.2–2.9) °	1.2 (0.1–4.3) °	3.8 (0.5–5.4) °	0.006
	MN, % (IQR)	0.0	0.0 (0.0–0.5)	0.5 (0.0–0.9)	0.7 (0.3–2.6) *	0.01
	Vacuoles, % (IQR)	0.0	0.0 (0.0–18)	0.7 (0.0–4.2)	0.0 (0.0–1.5)	ns
M	Binucleated, % (IQR)	4.0 (3.2–5.0)	1.4 (0.7–2.9)	2.0 (0.6–3.0)	0.9 (0.0–2.2) °	0.05
	MN, % (IQR)	0.0	0.0 (0.0–0.2)	0.0 (0.0–1.3)	0.0 (0.0–1.1)	ns
	Vacuoles, % (IQR)	1.6 (0.0–8.9)	1.4 (0.0–4.2)	2.1 (0.8–3.7)	0.0 (0.0–0.8)	ns

Data are expressed as the median and interquartile range (IQR). *p* calculated according to the Kruskal–Wallis test. Dunn’s post hoc test: * *p* < 0.01 vs. OA, ° *p* < 0.05 vs. OA. Abbreviations are as follows: OA, osteoarthritis; RA, rheumatoid arthritis; PsA, psoriatic arthritis; CIA, crystal-induced arthritis; PMN, polymorphonuclear cells; M, monocytes; L, lymphocytes; MN, micronucleus; IQR, interquartile range; ns, not significant.

**Table 3 ijms-24-05450-t003:** Cytokine, chemokine, and growth factor levels in SFs from OA (n = 7), RA (n = 6), PsA (n = 12), and CIA patients (n = 10).

	OA	RA	PsA	CIA	*p*
IL-1β, pg/mL (IQR)	0.4 (0.1–0.6)	10 (1.9–29) *	1.2 (0.8–2.3) °	13 (4.0–24) *	0.0001
IL-6, pg/mL (IQR)	90 (73–171)	4119 (182–13,852)	1070 (164–5043)	22,512 (2986–48,255) #	0.007
IL-8, pg/mL (IQR)	20 (6.9–41)	2872 (204–6095) #	222 (185–380)	934 (256–1815) #	0.001
IL-10, pg/mL (IQR)	1.0 (0.9–3.1)	30 (14–45) *	5.0 (3.3–10) §	15 (7.8–25) *	0.0001
TNF, pg/mL (IQR)	0.0	2.7 (0.3–9.8) #	0.0 (0.0–0.2) §	0.0 (0.0–4.2)	0.009
TGFβ, pg/mL (IQR)	1357 (875–2031)	2633 (1152–3317)	708 (412–1587)	1256 (652–3536)	ns

Data are expressed as the median and interquartile range (IQR) and *p* is calculated according to the Kruskal–Wallis test. Dunn’s post hoc test: # *p* < 0.01, * *p* < 0.001 vs. OA, ° *p* < 0.05 vs. CIA, § *p* < 0.05 vs. RA. Abbreviations are as follows: OA, osteoarthritis; RA, rheumatoid arthritis; PsA, psoriatic arthritis; CIA, crystal-induced arthritis; ns, not significant.

**Table 4 ijms-24-05450-t004:** Correlations between SF cytokine, chemokine, and growth factor levels and cellular inflammatory indices.

	WBC, n/mm^3^	PMN, %	M, %	L, %
IL-1β, pg/mL	*p* = 0.00001 (r = 0.84)	*p* = 0.0001 (r = 0.67)	*p* = 0.0001 (r = −0.64)	*p* = 0.002 (r = −0.51)
IL-6, pg/mL	*p* = 0.00001 (r = 0.79)	*p* = 0.00001 (r = 0.70)	*p* = 0.00001 (r = −0.67)	*p* = 0.0001 (r = −0.65)
IL-8, pg/mL	*p* = 0.0001 (r = 0.63)	*p* = 0.004 (r = 0.47)	*p* = 0.01 (r = −0.42)	Ns
IL-10, pg/mL	*p* = 0.00001 (r = 0.70)	*p* = 0.002 (r = 0.51)	*p* = 0.005 (r = −0.74)	*p* = 0.02 (r = −0.39)
TNF, pg/mL	ns	ns	ns	Ns
TGFβ, pg/mL	ns	ns	ns	Ns

Spearman’s rank correlation test. Abbreviations are as follows: WBC, white blood cell; PMN, polymorphonuclear cells; M, monocytes; L, lymphocytes.

**Table 5 ijms-24-05450-t005:** Correlations between the SF cytokine, chemokine, and growth factor levels and immature PMN, frequency of pyknosis, karyolysis, MN, and binucleated monocytes.

	PMN-Immature, %	Pyknosis, %	Karyoliysis, %	MN, %	M-Binucleated, %
WBC, n/mm^3^	*p* = 0.00001 (r = 0.78)	*p* = 0.001 (r = 0.52)	*p* = 0.03 (r = 0.38)	ns	*p* = 0.0001 (r = −0.61)
PMN, %	*p* = 0.00001 (r = 0.82)	*p* = 0.0001 (r = 0.63)	Ns	ns	*p* = 0.00001 (r = −0.70)
M, %	*p* = 0.00001 (r = −0.75)	*p* = 0.001 (r = −0.56)	Ns	ns	*p* = 0.00001 (r = 0.74)
L, %	*p* = 0.00001 (r = −0.73)	*p* = 0.001 (r = −0.54)	Ns	ns	*p* = 0.002 (r = 0.50)
IL-1β, pg/mL	*p* = 0.00003 (r = 0.64)	*p* = 0.003 (r = 0.49)	*p* = 0.004 (r = 0.48)	ns	*p* = 0.03 (r = −0.34)
IL-6, pg/mL	*p* = 0.00001 (r = 0.67)	ns	Ns	ns	*p* = 0.001 (r = −0.52)
IL-8, pg/mL	*p* = 0.001 (r = 0.53)	*p* = 0.00003 (r = 0.65)	*p* = 0.00001 (r = 0.69)	*p* = 0.05 (r = 0.35)	Ns
IL-10, pg/mL	*p* = 0.002 (r = 0.70)	*p* = 0.003 (r = 0.49)	*p* = 0.001 (r = 0.56)	ns	*p* = 0.02 (r = −0.38)
TNF, pg/mL	Ns	ns	*p* = 0.01 (r = 0.43)	ns	Ns
TGFβ, pg/mL	Ns	ns	*p* = 0.02 (r = 0.38)	ns	Ns

Spearman’s rank correlation test. Abbreviations are as follows: WBC, white blood cell; PMN, polymorphonuclear cells; M, monocytes; L, lymphocytes; MN, micronucleus.

**Table 6 ijms-24-05450-t006:** Correlations between cellular inflammatory indices, SF cytokines, chemokine, growth factor levels, and caspase-3 activity.

	CASPASE-3 Activity
WBC	*p* = 0.03 (r = 0.45)
PMN	*p* = 0.05 (r = 0.42)
M	Ns
L	*p* = 0.02 (r = −0.49)
IL-1β	*p* = 0.02 (r = 0.45)
IL-6	Ns
IL-8	*p* = 0.05 (r = 0.41)
IL-10	*p* = 0.02 (r = 0.47)
TNF	Ns
TGFβ	Ns

Correlations calculated using Spearman’s correlation test. Abbreviations are as follows: WBC, white blood cell; PMN, polymorphonuclear cells; M, monocytes; L, lymphocytes.

## Data Availability

Not applicable.

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
