# Peer review of "Leucocyte Abnormalities in Synovial Fluid of Degenerative and Inflammatory Arthropathies"

_ijms, 2023, doi:10.3390/ijms24065450_

Round 1

Reviewer 1 Report

The authors present a very interesting and complete manuscript where they assessed the percentage of cells´ populations with micronuclei and other genomic instability biomarkers in the synovial fluid of patients with different arthropathies. In order to give my full support for its publication in the journal I give the authors the following suggestions:

In the statistical analysis sub-heading of the method section the authors mention hat they used mean and standard deviation for reporting the data however in the tables they show interquartile range as dispersion measure and use non-parametric testing across the study. I encourage the authors to report the median as the central tendency measure for their data instead of the mean and correct the method section.

It is not clear if the synovial fluid obtained from patients with microcrystalline arthropathies were in the acute stage, and what type of crystal was present. It is relevant since the monosodium urate crystals induces in different grade the inflammatory response than pyrophosphate crystals.

It would be better if the authors present in tables or the text the actual spearman´s correlation coefficient along with the p value and not only the direction for the association to improve clarity o the study.

Since age is a mayor factor for genomic instability in cells of the immune system, I believe it would be necessary to do a multivariate analysis adjusting by the age of the patients to corroborate if the difference of the percentage of cells with micronuclei is not being masked by aging rather than the inflammatory process.  

Finally, the genetic expression analysis was carried out using total RNA extraction and it reflects the expression of all cells´ populations present in the synovial fluid of patients. This should be mentioned in the discussion section to make the interpretation of the results more straightforward.

Author Response

We thank the reviewer for her/his precious suggestions that will further improve our manuscript. We provide a point-by-point response to the reviewer’s comments (marked in red).

The authors present a very interesting and complete manuscript where they assessed the percentage of cells´ populations with micronuclei and other genomic instability biomarkers in the synovial fluid of patients with different arthropathies. In order to give my full support for its publication in the journal I give the authors the following suggestions:

  1. In the statistical analysis sub-heading of the method section the authors mention that they used mean and standard deviation for reporting the data however in the tables they show interquartile range as dispersion measure and use non-parametric testing across the study. I encourage the authors to report the median as the central tendency measure for their data instead of the mean and correct the method section. We modified the Tables 1, 2, 3 and the legends according to the Reviewer’ suggestions. Figures have been edited according to the Reviewer’ suggestions. We changed Figures 2, 3, 4, 5 and the legends. In Materials and Methods, we changed the text as follows: “Data are reported as median and IQR” (line 373).

  1. It is not clear if the synovial fluid obtained from patients with microcrystalline arthropathies were in the acute stage, and what type of crystal was present. It is relevant since the monosodium urate crystals induce in different grade the inflammatory response than pyrophosphate crystals. Thank you for this observation. Synovial fluids were obtained from patients with CIA in the acute stage and as we show in the graph below total and differential white blood cell count were not different between patients with gout or pseudogout. In the study we included 3 samples with MSU crystals and 7 samples with CPP crystals (we added this information in the method section, lines 306 - 307). Given the low number of samples we decided not to divide the samples according to the type of crystal. (see figure in the pdf file)
  2. It would be better if the authors present in tables or the text the actual spearman´s correlation coefficient along with the p value and not only the direction for the association to improve clarity of the study. We modified the Tables 4, 5, 6 and the legends as suggested.

  1. Since age is a major factor for genomic instability in cells of the immune system, I believe it would be necessary to do a multivariate analysis adjusting by the age of the patients to corroborate if the difference of the percentage of cells with micronuclei is not being masked by aging rather than the inflammatory process.  This is an important question. Definitely, age is an important factor influencing genomic instability, however, it is possible that a strong inflammatory milieu may be more genotoxic with respect to age. Furthermore, the oldest patients were in the non-inflammatory group (osteoarthritis). Performing a regression, the percentage of MN in our cohort of patients do not correlate with age, both considering the whole population or subdividing by disease.
  2. Finally, the genetic expression analysis was carried out using total RNA extraction and it reflects the expression of all cells´ populations present in the synovial fluid of patients. This should be mentioned in the discussion section to make the interpretation of the results more straightforward. Text has been modified according to the Reviewer’ suggestions at lines 269 - 270.

Reviewer 2 Report

The study also looked at the process of apoptosis (another form of programmed cell death) in these samples and found that the expression of BAX (a protein involved in apoptosis) was increased in CIA and RA compared to OA and PsA. On the other hand, the expression of Bcl-2 (an anti-apoptotic protein) was higher in CIA. caspase-3, an enzyme involved in the execution phase of apoptosis, was also found to be increased in SF of RA patients and associated with inflammatory and anti-inflammatory cytokines.

Taken together, these results suggest that inflammatory SF is associated with genomic instability and aberrant cell subpopulations. Further research is needed to fully understand the mechanisms behind these observations and their potential implications for the treatment of inflammatory arthropathies.

One of the biggest problems with this paper is that there is insufficient supporting evidence for the conclusions, and the authors should have added differential expression at the protein level rather than just looking at it with PCR. the PCR results assess changes at the mRNA level only, which is not accurate enough.

Another problem with this paper is that it does not fully explore the molecular mechanisms. The analysis of cytokines, local inflammatory cell indices, and apoptotic genes obtained by the authors was predictable. Is it then possible that other pathways are responsible for these differences? For example, endoplasmic reticulum stress? The authors did not investigate this either. In addition, apoptosis should be coupled with the results of flow cytometry analysis to strengthen the credibility of the work.

Evaluation at the cellular level would make the results of this paper more heterogeneous, which affects the cytological evaluation of leukocytes and weakens the reproducibility of the results of this paper. How did the authors address this issue?

Author Response

We thank the reviewer for her/his precious suggestions that will further improve our manuscript. We provide a point-by-point response to the reviewer’s comments (marked in red).

The study also looked at the process of apoptosis (another form of programmed cell death) in these samples and found that the expression of BAX (a protein involved in apoptosis) was increased in CIA and RA compared to OA and PsA. On the other hand, the expression of Bcl-2 (an anti-apoptotic protein) was higher in CIA. caspase-3, an enzyme involved in the execution phase of apoptosis, was also found to be increased in SF of RA patients and associated with inflammatory and anti-inflammatory cytokines. Taken together, these results suggest that inflammatory SF is associated with genomic instability and aberrant cell subpopulations. Further research is needed to fully understand the mechanisms behind these observations and their potential implications for the treatment of inflammatory arthropathies.

  1. One of the biggest problems with this paper is that there is insufficient supporting evidence for the conclusions, and the authors should have added differential expression at the protein level rather than just looking at it with PCR. The PCR results assess changes at the mRNA level only, which is not accurate enough.

We thank the reviewer for this important observation. Indeed, protein level should be determined to support the PCR analysis. We evaluated by western blot the proteins Bax and Bcl-2 on which we reported the most significant variations. The analysis was carried out on synovial fluids analyzed by PCR except for one OA fluid of which we won’t have any more lisate to use. We reported the results in the supplementary file, integrating their description in the text.

  1. Another problem with this paper is that it does not fully explore the molecular mechanisms. The analysis of cytokines, local inflammatory cell indices, and apoptotic genes obtained by the authors was predictable. Is it then possible that other pathways are responsible for these differences? For example, endoplasmic reticulum stress?

We agree with the reviewer that additional data on endoplasmic reticulum stress would add value to the conclusions. Unfortunately, the volume of SF collected was very variable between patients ranging from a few millilitres to maximum 10-12 ml and in case we perform WB to assess PCR results (query N.1), we won’t have any more lisate to use. Furthermore, we couldn’t use samples from new patients as it is essential to compare factors and pathways in the same samples. This mechanism is indeed very interesting in terms of unfolded protein response, and we will definitely consider this aspect for future studies.

The authors did not investigate this either. In addition, apoptosis should be coupled with the results of flow cytometry analysis to strengthen the credibility of the work.

We agree with the reviewer that results of flow cytometry analysis to strengthen the credibility of the work, however this type of analysis should be conducted on freshly extracted synovial fluid that are not available.

  1. Evaluation at the cellular level would make the results of this paper more heterogeneous, which affects the cytological evaluation of leukocytes and weakens the reproducibility of the results of this paper. How did the authors address this issue?

As specified above, we agree and we performed additional analysis on Bax and Bcl-2 protein levels, but they were limited by the amount of the samples.

Reviewer 3 Report

Baggio et al., in the manuscript entitled “Leucocyte abnormalities in synovial fluid of degenerative and inflammatory arthropathies”, discussed in synovial fluid, the relation between inflammatory and genomic instability / cell abnormalities of rheumatoid arthritis, psoriatic arthritis, crystal-induced arthritis, and osteoarthritis.

Overall, this paper is well-designed and structured. There are several issues as follows:

1. Why the authors chose RA, PsA, and CIA to represent inflammatory arthropathies, and OA for non-inflammatory arthropathies? Are they really representative for inflammatory and genomic instability?

2. Since the authors only chose 3 (RA, PsA, and CIA), I am not quite sure that the conclusion "inflammatory SF 26 are associated with genomic instability and abnormal cell subsets" is valid. Maybe CIA is a special case to show the association? Any way to exclude that the possibility that CIA is just a special case? 

3. What is the difference between MN in table 2 and figure 2. It seems that they are not consistent?

4. Why the authors sometimes use table, sometimes use figure to show the results? It is better to also provide a table of values of each figure.

Author Response

We thank the reviewer for her/his precious suggestions that will further improve our manuscript. We provide a point-by-point response to the reviewer’s comments (marked in red).

Baggio et al., in the manuscript entitled “Leucocyte abnormalities in synovial fluid of degenerative and inflammatory arthropathies”, discussed in synovial fluid, the relation between inflammatory and genomic instability / cell abnormalities of rheumatoid arthritis, psoriatic arthritis, crystal-induced arthritis, and osteoarthritis. Overall, this paper is well-designed and structured. There are several issues as follows:

  1. Why the authors chose RA, PsA, and CIA to represent inflammatory arthropathies, and OA for non-inflammatory arthropathies? RA, PsA and CIA are the most prevalent inflammatory arthropathies in adults, characterized by high number of WBC and PMN%. Osteoarthritis is a non-inflammatory degenerative arthropathy and, although a subclinical inflammation has been demonstrated in the progression of the disease, SFs present less than 500 cells/mm3 and classified as non-inflammatory (Schumacher, H.R., Reginato, A.J. Atlas of Synovial Fluid Analysis and Crystal Identification,1991; Brian F. Synovial Fluid Analysis and The Evaluation of Patients with Arthritis, Mandell Editor, 2022). Are they really representative for inflammatory and genomic instability? Regarding the genomic instability “Recent studies suggest that some rheumatological diseases, including rheumatoid arthritis, are associated with overall genomic instability in the T cell compartment and increased sensitivity to DNA damage. However, no data regarding leucocytes abnormalities in synovial fluid and their relationship with inflammation are available. This is reported in the introduction section.

  1. Since the authors only chose 3 (RA, PsA, and CIA), I am not quite sure that the conclusion "inflammatory SF 26 are associated with genomic instability and abnormal cell subsets" is valid. We thank the reviewer for this observation. Actually RA, PsA and CIA are the most common inflammatory arthropathies providing classical inflammatory effusions. Maybe CIA is a special case to show the association? Any way to exclude that the possibility that CIA is just a special case? This is an important point to consider deeper as pathogenic crystals may indeed induce genomic instability through ROS induction and cytokine release. We discussed this aspect in the discussion section, but further studies are needed to address this issue.

  1. What is the difference between MN in table 2 and figure 2. It seems that they are not consistent? We have reported in the table the micronuclei present in monocytes and neutrophils while the figure shows the total micronuclei in monocytes and neutrophils.

  1. Why the authors sometimes use table, sometimes use figure to show the results? It is better to also provide a table of values of each figure. We thank the reviewer for this observation. Due to large number of figures and tables we added a supplementary file with new tables and the legends as suggested (Tables 7, 8, 9, 10).

see pdf file

Round 2

Reviewer 3 Report

I have no further comments.

Author Response

We thank the reviewer for all the comments. We submitted a second revision of the manuscript to accomodate all reviewers' queries.